# Plasma Cell-Free Tumor Methylome as a Biomarker in Solid Tumors: Biology and Applications

Danielle Benedict Sacdalan [1,2] , Sami Ul Haq [2,3] and Benjamin H. Lok [1,2,4,*]

1 Institute of Medical Science, Temerty Faculty of Medicine, University of Toronto, 1 King's College Circle, Medical Sciences Building, Room 2374, Toronto, ON M5S 1A8, Canada
2 Radiation Medicine Program, Princess Margaret Cancer Centre, 610 University Ave, Toronto, ON M5G 2C4, Canada
3 Schulich School of Medicine & Dentistry, Western University, 1151 Richmond St, London, ON N6A 5C1, Canada
4 Department of Medical Biophysics, Temerty Faculty of Medicine, University of Toronto, 101 College Street, Room 15-701, Toronto, ON M5G 1L7, Canada
* Correspondence: benjamin.lok@uhn.ca

**Abstract:** DNA methylation is a fundamental mechanism of epigenetic control in cells and its dysregulation is strongly implicated in cancer development. Cancers possess an extensively hypomethylated genome with focal regions of hypermethylation at CPG islands. Due to the highly conserved nature of cancer-specific methylation, its detection in cell-free DNA in plasma using liquid biopsies constitutes an area of interest in biomarker research. The advent of next-generation sequencing and newer computational technologies have allowed for the development of diagnostic and prognostic biomarkers that utilize methylation profiling to diagnose disease and stratify risk. Methylome-based predictive biomarkers can determine the response to anti-cancer therapy. An additional emerging application of these biomarkers is in minimal residual disease monitoring. Several key challenges need to be addressed before cfDNA-based methylation biomarkers become fully integrated into practice. The first relates to the biology and stability of cfDNA. The second concerns the clinical validity and generalizability of methylation-based assays, many of which are cancer type-specific. The third involves their practicability, which is a stumbling block for translating technologies from bench to clinic. Future work on developing pan-cancer assays with their respective validities confirmed using well-designed, prospective clinical trials is crucial in pushing for the greater use of these tools in oncology.

**Keywords:** DNA methylation; liquid biopsy; biomarkers; cell-free DNA; 5-methylcytosine





## 1. Introduction

DNA methylation is a fundamental mechanism of epigenetic control in cells and its dysregulation is strongly implicated in cancer development. The addition of a methyl group to cytosine, particularly at CpG islands (CGIs) results in changes in gene expression and affects the transcription of genomic repeats and non-coding RNA. These result in changes within cancer cells as well as the tumor microenvironment (TME) that favor the development of disease. Patterns of tumor-specific methylation are recapitulated in cell-free DNA (cfDNA) allowing use of the latter for the study of cancer epigenetics. Liquid biopsies can be used to obtain cell-free DNA from different biofluids such as cerebrospinal fluid, pleural effusion, urine, and plasma. Of the different sources of cfDNA, plasma has seen extensive use as a source of methylated DNA for use in biomarker studies. Diagnostic, prognostic, predictive, and treatment response monitoring applications have or are being developed around methylated cfDNA obtained from plasma.

This narrative review aims to provide the reader with an overview of the biology that underpins the use role of DNA methylation in cancer. It will then discuss the utility of

cfDNA methylation as a cancer biomarker. Finally, it will outline the different opportunities and challenges of using cfDNA methylation in oncology. The methodology and search strategies employed in writing this work are summarized in Appendix A.

## 2. DNA Methylation in Cancer

### 2.1. Biology of DNA Methylation in Malignancy

2.1.1. Aberrant DNA Methylation of CpG Islands Are Seen in Cancer

More than half of all human promoters overlap with 1–2 kb CG-rich regions, referred to as CpG islands (CGIs). Nearly 95% of CGI promoter regions are unmethylated in normal tissues. While CGIs are preferentially found near promoters and transcription start sites (TSS), they may also appear within gene regulatory elements, gene bodies, and intergenic regions [1]. Cancers typically display focal regions of tumor-specific hypermethylation, especially at CGIs, against the backdrop of global hypomethylation that impacts gene expression and carcinogenesis [2–4].

2.1.2. Gene Promoter Hypermethylation Leads to Silencing of Tumor Suppressor Genes

Hypermethylation of transcriptional regulatory regions of tumor suppressor genes (TSGs) results in their silencing and the abrogation of their anti-tumor function [2,5]. First described in colorectal cancer (CRC), the CpG island methylator phenotype (CIMP) that is characterized by the expansive TSG promoter silencing has since been documented in other solid malignancies [6]. DNA hypermethylation was observed in promoters of classic TSGs such as retinoblastoma transcriptional corepressor 1 (*RB1*) and cell cycle regulators *CDKN2A*, among many others [1,2].

Koudonas et al. [7] demonstrated that hypermethylation of the TSGs *PCDH17*, *NEFH*, and *RASSF1A* in resected renal cell carcinoma compared to normal tissue was associated with worse survival outcomes. A 2020 study of paired tumor tissue samples and matched healthy control tissue from patients with prostate cancer identified 7500 differentially methylated regions (DMRs). Compared to matched healthy controls, prostate cancer tissue was hypermethylated in promoter regions of well-known tumor-suppressor genes such as *CDKN2A*, resulting in silencing [8]. Promoter methylation is also responsible for the silencing of genes whose loss of function leads to invasion and metastasis, such as the cell adhesion regulators E-cadherin (*CDH1*) and H-cadherin (*CDH13*) [9,10]. Moreover, Chen et al. [11] profiled the cell-free methylomes of 235 early and advanced prostate cancers and found that promoters of TSGs show significantly higher methylation in metastatic castration-resistant tumors compared to localized castration-naïve disease, suggesting that TSG promoter hypermethylation not only plays a role in carcinogenesis but also in disease progression.

2.1.3. Methylation Can Lead to Epigenetic Activation of Oncogenic Pathways

Hypomethylation of oncogene promoters typically results in their expression. Oncogene products dysregulate cell proliferation, migration, and invasion. This results in elevated metastatic potential, increased stemness in cells, or immune evasion [12]. For example, BCL6 acts as a transcriptional regulator that represses tumor suppressors thereby favoring cell proliferation and survival [13]. Expression of BCL6 is epigenetically regulated and knockdown of UHRF1, an upstream regulator, results in loss of *BCL6* gene promoter methylation thereby favoring its expression. Notably, BCL6 was observed to be overexpressed in exhausted T-cells and may play a role in tumor immune escape [14]. In addition to promoters, hypomethylation of other gene regulatory regions, such as enhancers, was shown to correlate with the expression of cancer-specific genes and pathways across multiple cancers [15,16].

In contrast to promoter hypermethylation, gene body hypermethylation is posited to correlate positively with gene expression [1,17,18]. As an example, the homeobox superfamily of transcription factors is essential to the regulation of cell growth and differentiation. These genes have high CpG density, which makes their expression susceptible to epigenetic

control [19]. In relation to this, Su et al. [20] reported that gene-body hypermethylation results in the overexpression of approximately 43% of known homeobox genes, many of which are associated with oncogenesis.

### 2.1.4. Global DNA Hypomethylation Leads to Chromosomal Instability

DNA methylation plays a critical role in maintaining genomic integrity by regulating the transcription of constitutive heterochromatin elements, sub-telomeric regions, and genomic repeats [21,22]. Expansive hypomethylation is common in cancer and may involve regions as large as ten megabases in size, which can render the cancer genome susceptible to genomic instability. Moreover, increased gene transcription from loss of gene promoter methylation results in the formation of DNA-RNA R-loops that can cause replication stress and DNA damage—further exacerbating genome instability [21,23]. Reduced methylation destabilizes heterochromatin at pericentromeric regions and is associated with loss of tumor suppression and oncogenesis in gastrointestinal cancers and sarcomas [21,22]. Telomere length is affected by methylation of sub-telomeric regions, and its dysregulation is considered a cancer hallmark [21,24,25].

### 2.1.5. DNA Methylation Influences Gene Expression beyond Promoter Silencing

Alternative mRNA splicing is responsible for transcriptional diversity in cells [26,27]. During splicing, non-coding introns are removed, and coding exons are ligated to form mature mRNA at specific positions called splice sites. Recognition of introns and exons by the spliceosome machinery is critical for the correct execution of this process [26]. DNA methylation modifies the accessibility of exonic nucleosomes to RNA Polymerase II, which demonstrates the regulatory role it plays in splicing [26,27].

Chen and Elnitski [28] showed that methylation-correlated isoforms affect the functional protein domains of gene products. Moreover, these gene sets were enriched for oncogenes, tumor suppressors, and cancer-related pathways across multiple cancers [28].

### 2.1.6. Methylation Can Influence the Non-Coding Genome, Leading to Cancer

DNA hypomethylation led to the expression of genomic repeat elements and transposable elements that interfered with chromosomal integrity and drove oncogene expression in a process called onco-exaptation [21,22]. Jang et al. [29] reported RNA-seq data from 7769 tumors across 15 cancer types from the TCGA. In this cohort, the expression of 106 oncogenes was regulated by onco-exaptation in nearly half (49.7%) of tumors. To validate this relationship, the authors performed in vitro induction of DNA methylation on the promoter of the short-interspersed nuclear element (SINE) *AluJb* by CRISPR. This resulted in a 20–30% increase in methylation and consequently a 40% reduction in the expression of the oncogene *LIN28B*. In addition to onco-exaptation, transposon-associated dsRNA-dependent paracrine signaling had been found to be favorable to tumor growth in breast, lung, and pancreatic cancer [30,31].

The study of non-coding RNA is a growing area of interest owing to the emerging roles that it plays in disease [32,33]. The interaction between non-coding RNA (ncRNA) and DNA methylation is complex. Few studies have comprehensively examined if these non-coding regions are methylated, with most investigations focusing on a specific type of ncRNA called long non-coding RNA (lncRNA).

Yang et al. [34] showed that DNA methylation worked in concert with lncRNA to regulate the expression of protein-coding genes across 18 different cancer types. Many lncRNAs interacted with the histone-lysine N-methyltransferase EZH2 and DNA methyltransferases (DNMT) in both cancer-specific and pan-cancer patterns to methylate DNA downstream of target genes. The interplay of methylation writers and erasers governed the expression of specific lncRNAs and resulted in the regulation of distinct cell signaling pathways that favored cancer progression. For example, Li et al. [35] demonstrated that *LINC01270* interacted with DNMTs to hypermethylate the *LAMA2* promoter region, suppressing the negative regulatory effects of the latter on MAPK signaling and leading to breast cancer

progression. These effects were then reversed by RNAi knockdown of *LINC01270*. Another lncRNA, MAGI2-AS3 modulated the Wnt/β-catenin signaling pathway and decreased tumorigenicity in breast cancer by downregulating methylation of the *MAGI2* gene promoter. This tumor suppressive effect was countered by the activity of TET1 on MAGI2-AS3, which demonstrated epigenetic control of TSG function [36].

Several comprehensive reviews discuss the association between DNA methylation and lncRNA expression in cancer [37,38]. In contrast to lncRNAs, other types of noncoding RNA have not been examined as extensively in the context of DNA methylation. This underscores the complexity of the relationship between ncRNA and DNA methylation in cancer [39].

### 2.1.7. Role of DNA Methylation in the Tumor Microenvironment

Methylation dynamics play a key role in contextualizing the tumor microenvironment (TME). To illustrate, TET2 demethylates promoters of genes encoding cytokines and transcription factors that influence CD4$^+$ T-cell fate. Expression of the cytokine IL-4 favors differentiation into Th2 cells while expression of the transcriptional regulator FOXP3 expression leads to differentiation into tissue regulatory T-cells (Treg) [40]. In CD8$^+$ cells, active demethylation of gene enhancer regions by TET2 facilitates differentiation to T-effector cells. Conversely, the conditional loss of TET activity was shown to drive a shift towards a memory T-cell fate [40–42]. Meanwhile, DNMT3A-directed de novo methylation of the promoter of T-cell-specific transcription factor 7 (*Tcf7*) results in the suppression of memory T-cell differentiation and supports differentiation into effector subtypes [41,42].

As with T-cells, myeloid differentiation is epigenetically influenced by DNA methylation. TET2 acts in concert with IL-4 and STAT6 to promote the expression of ITGB2, which is a cell adhesion molecule important in monocyte-to-dendritic cell differentiation and function. Dendritic cells have a central role in antigen presentation and T-cell activation within the TME [40]. Myeloid-derived suppressor cells (MDSC) are key immune regulators in the TME that serve a pro-tumorigenic function by promoting immune tolerance to cancer. An increase in DNMT3A expression in MDSCs in the presence of Prostaglandin E2 results in a gain of DNA methylation and silencing of immunogenicity-associated genes (e.g., *FAS, RUNX1*, and *S1PR4*) [43,44]. Finally, macrophage polarization towards either the activated (M1) phenotype or the inhibitory (M2) phenotype is also subject to the influence of DNA methylation. Express of DNMT3b and DNMT1 are associated with M1-like macrophage polarization through its effect on PPAR γ1 [45]. Meanwhile, siRNA knockdown of the *DNMT3B* gene increases M2 marker expression in macrophages. A similar increase in M2 markers was seen following the treatment of M1 macrophages with DNMT inhibitors (DNMTi) [45–47].

The development of immune escape mechanisms that bar cytotoxic effector cells from infiltrating the TME is key to tumor progression [25]. DNA methylation also plays a part in this process. DNMT1 activity was shown to disturb the trafficking of CD8$^+$ T-cells into the TME of a murine ovarian cancer cell model by repressing the tumor production of CXCL9 and CXCL10 [40]. In addition, tumors use DNA methylation to negatively regulate chemokines such as CCL5 and CCL2, which are essential in T-cell and macrophage chemotaxis. Epigenetic silencing of the *CCL2* gene by methylation was found to diminish macrophage infiltration and promote disease progression in small-cell lung cancer (SCLC) [48–50]. As described previously, DNA methylation also plays a role in the activation of inhibitory immune cells such as MDSC that modulate a pro-tumor TME. Interestingly, MDSC infiltration can be reduced by treatment with DNMTi [51].

Investigations into the mechanisms behind effector T-cell exhaustion revealed the contribution of multiple epigenetic mechanisms including DNA methylation. De novo methylation by DNMT3A established a stable exhausted state and its deletion in CAR T-cells as shown by Prinzing et al. [52] led to a more anti-tumor state. Moreover, DNA methylation profiling of exhausted tumor-infiltrating lymphocytes (TIL) identified promoter hypomethylation of T-cell exhaustion markers PD-1 and HAVCR2. Binding sites

of exhaustion-associated transcription factors such as NR4A1 were also seen to be hypomethylated, implying that activation of the T-cell inhibitory programming was governed, at least in part, by methylation [53]. Taken together, these highlight the complex roles DNA methylation plays in the tumor TME.

Figure 1 summarizes the disparate roles DNA methylation plays in the development of cancer.

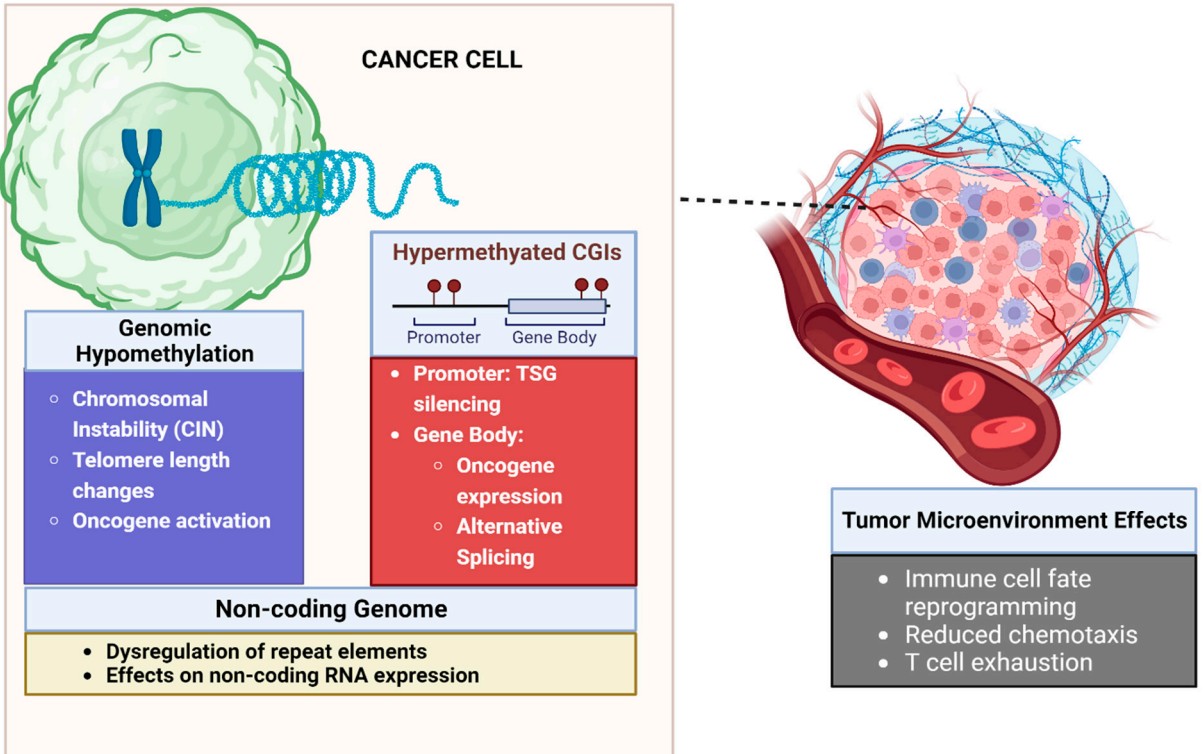

**Figure 1.** Influence of DNA methylation on the tumor and tumor microenvironment. Cancers are characterized by extensive genomic hypomethylation that leads to chromosomal instability and oncogene activation. Focal hypermethylation of CpG islands at promoters leads to TSG silencing. Hypermethylation of gene bodies can lead to oncogene expression and alternative splicing leading to cancer-related isoforms of genes. In the tumor microenvironment DNA methylation can lead to altered immune cell fate and trafficking. DNA methylation also underpins epigenetic mechanisms of T-cell exhaustion. Created with Biorender.

### 2.2. Cell-Free Tumor Methylome Recapitulates the Cancer Epigenome

Cell-free DNA (cfDNA) refers to DNA fragments in the noncellular component of blood. Although it has been an area of great interest over the last decade, Mandel and Métais were the first to demonstrate DNA in the blood of both healthy and diseased patients in 1948 [54]. cfDNA arises from cells through necrosis and apoptosis, but also enters the circulation via active secretion [54,55]. Circulating DNA from normal cells is found in low concentrations in plasma (~10–15 ng/mL) but can increase due to acute stressors like disease and injury [56]. Notably, it was shown that levels of cfDNA in individuals with cancer are higher in comparison with their counterparts without cancer [56]. The component of cfDNA that originates from tumors is referred to as circulating tumor DNA (ctDNA) and represents a smaller fraction of overall cfDNA [57,58]. These circulating tumor fragments are shorter than DNA from non-cancer cells and range in size from ~140 to 200 bp in length, corresponding to the size of nucleosome-associated DNA [54,59]. The ctDNA fraction in plasma varies but increases in association with disease type and burden. Advanced disease and high-shedding tumors such as SCLC and prostate cancer may have fractions > 20% of total cfDNA [60–63]. Conversely, reduced ctDNA fractions in the blood are detectable

in early-stage disease or following effective anticancer therapy [56,60,62,64]. In addition to plasma, ctDNA can be obtained from other biofluids such as cerebrospinal fluid, saliva, pleural effusions, urine, ascites, and stool representing multiple sources of nucleic acid for study [56]. Depending on the source, ctDNA size may vary because of disparate conditions within the respective biological compartments from which the nucleic acids are sourced. For instance, shorter ctDNA fragments may be observed in urine compared to plasma owing to the greater nuclease activity in the former [54].

While the term liquid biopsy is most often used to refer to the analysis of cfDNA from peripheral blood, it also pertains to the isolation and study of tumor-derived material from other bodily fluids. Liquid biopsies using standard venipuncture afford several distinct advantages over conventional tissue sampling. First, their minimally invasive nature allows for evaluation of the disease, particularly in cases where tumor access is difficult, unsafe, or ethically challenging, especially for repeated longitudinal sampling [56]. Moreover, in addition to ctDNA, circulating tumor microRNA, extracellular vesicles, tumor-associated proteins, and circulating tumor cells (CTCs) can also be sampled at the point of care by this approach [65]. From a testing standpoint, utilizing pooled cfDNA obtained from liquid biopsies to determine a positive signal—such as the presence of cancer in a patient—avoids dependence on specific fragments and thereby improves sensitivity. This is especially important when dealing with scenarios wherein the amount of input DNA is constrained, and adequate representation of individual DNA fragments cannot be guaranteed [66]. Another advantage of liquid biopsies is their ability to better recapitulate the molecular heterogeneity harbored by multiple distinct clonal populations that collectively shed cfDNA into the blood. This is because the molecular diversity of disease is not as easily captured by needle aspirates of single tumor sites and multiple needle passes to sample different sites may be difficult or excessively invasive [56]. Finally, from a therapeutic perspective, this technique allows for longitudinal sampling across successive time points providing opportunities for studying tumor evolution resulting from treatment [56,58].

Early applications of liquid biopsy technology involve genotyping and mutation detection, but indications for its use have since expanded to include epigenetic analyses, including the study of ctDNA methylation [58,67–69]. The study of ctDNA methylation in turn has several advantages. First, cancer-specific methylation changes occur earlier in tumorigenesis and represent an opportunity for early disease detection and diagnosis [57,70]. Moreover, combining methylation with orthogonal methods such as mutation detection and fragmentomics to profile ctDNA further enhances the discriminative power of such tests by interrogating biologic features that are characteristic of tumor-derived DNA [71]. Second, methylation patterns in ctDNA are consistent with their tissue-of-origin, allowing for the location of the source of a cancer, especially those with an unknown primary site [68,72]. Third, tumor cells tend to exhibit more homogenous DNA methylation changes versus gene mutations that tend to exhibit greater intra-tumoral heterogeneity [67]. The consistency of the methylation signal opens the door for evaluating tumor evolution, especially after being subjected to the selective pressure of anti-cancer therapy. In the context of treatment resistance, examining changes in differentially methylated regions (DMR) in ctDNA obtained from longitudinal blood draws can shed light on epigenetic mechanisms that underpin treatment resistance. Various bisulfite conversion (e.g., quantitative methylation-specific PCR and cfMethyl-seq) and antibody-enrichment-based (e.g., cfMeDIP-seq) platforms for the study of ctDNA methylation have been developed in the last decade. The details of these assays are beyond the scope of this review but were extensively discussed elsewhere [73–76].

### 2.3. Applications of cfDNA Methylation as Biomarker in Cancer

#### 2.3.1. Methylation as a Diagnostic Biomarker

The concordance between tumor and cell-free cancer methylomes means that it can be leveraged as a tool for cancer diagnosis. Assays that employ ctDNA methylation for this application vary in their approaches. Some use targeted detection of methylated sites

in specific genes such as *SEPT9* [77,78], *SHOX2* [79], *PTGER4* [79], and *SDC2* [80] while others profile more extensive swathes of the genome.

Classifiers based on more extensive DNA methylation profiles to detect cancers via plasma-based liquid biopsies were also developed. In this regard, the most significant application of ctDNA methylome profiling is in multicancer early detection (MCED). The importance of this application lies in the fact that early diagnosis of disease allows for the administration of effective, curative treatment to optimize patient outcomes [81]. The results of the Circulating Cell-free Genome Atlas (CCGA) study (NCT02889978) demonstrated that ctDNA methylation profiling using the GRAIL test could detect cancer signals across >50 cancer types with an overall accuracy of predicting cancer signal origin (CSO), i.e., the tissue of origin of cancer in cases of true positives is 88.7% (95% CI: 87.0–90.2%). This study showed that interrogation of ctDNA methylation could outperform tests that relied on copy number and structural variants for cancer detection [82]. Results from the prospective SYMPLIFY (ISRCTN10226380) and PATHFINDER (NCT04241796) cohort studies further suggested the feasibility of this approach. In SYMPLIFY, 6238 symptomatic individuals with possible underlying cancer as a cause were recruited and followed until diagnostic resolution or nine months. The primary study outcome was the diagnostic performance of the MCED test for the detection of new diseases. The specificity of the test was 98.4% (95% CI: 98.1–98.8%), while the sensitivity was 66.3% (95% CI: 61.2–71.1%). For cases wherein a cancer signal was correctly detected, the accuracy for predicting the CSO was 85.2% (95% CI: 79.8–89.3%) [83]. Meanwhile, the PATHFINDER cohort included individuals 6662 aged 50 and older regardless of their cancer risk factors, who were not being evaluated for or had any history of cancer. The primary objective was the determination of the time needed to achieve diagnostic resolution in individuals with a positive MCED test. A cancer signal was detected in 92 (1.4%) participants, of whom 35 (38%) are true positives and 57 (62%) are false positives. The time to diagnostic resolution was found to be 57 days (95% CI: 33–143) in true-positive participants and 162 days (95% CI: 44–248) in false-positive participants. False positive results were a challenge for screening tests as individuals with this result needed to undergo further diagnostic testing. In this study among those with false positive results, <1% underwent invasive surgical or non-surgical procedures. PATHFINDER likewise assessed the performance of GRAIL and found that the test specificity was 99.1% while the sensitivity to detect disease over the one-year follow-up period was 29%. Notably, the test could accurately predict the CSO in 97% of true positives [84]. Additional clinical trials were ongoing to validate the utility of ctDNA methylation profiling for MCED: STRIVE (NCT03085888), SUMMIT (NCT03934866), NHS-GALLERI (ISRCTN91431511), and PATHFINDER2 (NCT05155605) [82,85–87].

A second platform, this time that leveraged cfMeDIP-seq, an antibody-based, non-degradative genome-wide DNA methylation enrichment assay was being developed for use in MCED. Validation of this test was performed on a cohort of 4332 consisting of untreated patients across eight different tumor types—bladder, breast, CRC, head, and neck (HNSCC), lung, ovarian, prostate, and renal cell (RCC) cancers, and a set of age and sex-matched controls [88]. The test performed well with an AUROC of 0.94 (95% CI: 0.93, 0.96) across all tumors and was robust in each individual tumor type including low cfDNA shedding tumors—bladder, breast, prostate, and RCC—with an AUROC 0.92 (0.91, 0.94) [88]. The high detection of low-shedding disease was particularly noteworthy since early-stage cancers with low tumor burden released small amounts of cfDNA into the circulation and were an important hurdle to overcome for viable MCED tests.

In patients diagnosed with cancer, methylation profiling of ctDNA also showed utility in tumor subtyping. Chemi et al. [89] reported that tumor-specific methylation patterns could discriminate between transcription factor-defined SCLC subtypes. This reflected the association between methylation and molecular heterogeneity within cancer types.

Table 1 outlines examples of plasma cfDNA methylation-based diagnostic biomarkers in different stages of development.

**Table 1.** Plasma cfDNA methylation-based diagnostic biomarkers in clinical use or development.

| Application | Cancer | Test/Assay/Target | Study Size (N) | Sensitivity | Specificity | Comments | References |
|---|---|---|---|---|---|---|---|
| Diagnosis | Colon | Epi proColon® / methylation specific PCR/*SEPT9* | 245 | 67% (95% CI: 56–77%) | 89% (95% CI: 83–93%) | Obtained FDA approval in 2016 | [77] |
| Diagnosis | Lung | Epi proLung® / methylation specific PCR/*SHOX2* and *PTGER4* | 172 | AUROC = 0.88; at specificity = 90%, sensitivity = 67%; at sensitivity = 90%, specificity = 73%. | | Obtained CE-IVD mark in 2017 | [79] |
| Diagnosis | Colon | EarlyTect® Colon/methylation-specific PCR/*SCD2* | 256 | 87.0% (95% CI: 80.0–92.3%) | 95.2% (95% CI: 89.8–98.2%) | -- | [80] |
| Diagnosis | Lung | EarlyTect® Lung Cancer/methylation-specific PCR/*PCDHGA12* | -- | 75.0% (95% CI: 61.8–81.8%) | 78.9% (95% CI: 62.2–89.8%) | In development for use with bronchial washings or blood | [90] |
| Diagnosis (MCED) | Pan-cancer | GRAIL(Galleri™)/ NGS and custom classifier/ methylation panel covering $1.1 \times 10^6$ CpGs | 4077 | 51.5% (95% CI: 49.6–53.3%) | 99.5% (95% CI: 99.0–99.8%) | Data based on CCGA trial (NCT02889978) | [91] |
| MRD | Pan-cancer | Guardant Reveal™/NGS and custom classifier/ 500 genes and 4Mb of DMRs | CRC cohort = 84; Breast cancer cohort = 20 | CRC cohort: 55.6% (95% CI: 35.3–74.5) Breast cancer cohort: sensitivity = 85% [1] | CRC cohort: 100% (95% CI: 90.5–100) Breast cancer cohort: specificity = 100% [1] | -- | [92,93] |
| Diagnosis (MCED)/ Prognosis/ MRD | Pan-cancer | ADELA/cfMeDIP-seq/whole genome | MCED = 4322 Prognosis: HNSCC cohort = 93; RCC cohort = 151 | MCED: Multi-cancer cohort—Cancer cases discriminated from controls with an AUROC of 0.94 (95% CI: 0.93, 0.96). In low cfDNA shedding tumors, AUROC was 0.92 (95% CI: 0.91, 0.94). Prognosis: HNSCC—Likelihood of recurrence or progression (HR 3.51, 95% CI: 1.1–11.19, $p = 0.034$); RCC—Likelihood of recurrence or progression (HR 13.28, 95% CI: 5.47–32.26, $p < 0.001$) | | MRD assay currently in development | [88,94,95] |

[1] 95% CI: not reported.

### 2.3.2. Methylation as a Prognostic Biomarker of Patient Outcomes

As a ubiquitous mode of epigenetic regulation, DNA methylation of specific genes provides value in prognosticating outcomes in cancer. Huang et al. [96] utilized bisulfite conversion-based real-time PCR to determine the methylation status of the SEPT9 gene (*mSEPT9*) in the plasma of 144 preoperative CRC patients. Patients who were *mSEPT9*+ had lower disease-free survival (DFS) rates than those who were *mSEPT9*− (two-year DFS: 52.1% vs. 73.9%, $p = 0.014$), Moreover, *mSEPT9* was an independent predictor of prognosis (HR = 2.741, $p = 0.009$) in multivariate regression analysis [96]. Of interest, a 2021 systematic review by Hier et al. [97] identified over 100 different genes regulated by methylation that carried prognostic significance. Apart from protein-coding genes, the loss of methylation of repeat elements was significant in early carcinogenesis and correlated with poor prognostic outcomes [98,99]. Chen et al. [100] studied the methylation status of the *Alu* element in cfDNA from 109 glioma patients. The authors found that *Alu* methylation was lower in glioma samples versus non-cancer controls ($p < 0.01$) and that methylation of *Alu*

correlated negatively with disease severity—high-grade gliomas showed less methylation than low-grade gliomas ($p < 0.01$). Conversely, *Alu* methylation correlated positively with survival ($p < 0.01$).

Recognizing that the tumor methylation signal was distinct from that of the non-cancer genome, Ul Haq et al. [101] performed a whole genome approach to the assessment of the cell-free methylome of SCLC using cfMeDIP-seq. The authors utilized an in silico approach using paired peripheral blood leukocytes to subtract non-cancer noise from cfDNA methylation data similar to previous studies [71]. This approach identified two biologically distinct methylation-defined SCLC prognostic groups with differences in overall survival [101]. Validation of ctDNA methylation as a prognostic biomarker in HNSCC and RCC was performed and data from these studies were presented in abstract form at the ESMO 2023 conference (Table 1) [94,95].

2.3.3. Methylation as a Predictive Biomarker of Treatment Response

Methylation of the O$^6$-alkylguanine DNA alkyltransferase (*MGMT*) promoter had been a known predictor of response to the alkylator temozolomide in glioma [65]. The promoter methylation status of this gene could be assessed using ctDNA obtained from blood or cerebrospinal fluid via targeted, methylation-specific PCR and was found to be highly concordant with tumor tissue [102]. Meanwhile, the expression of the DNA/RNA helicase Schlafen 11 (SLFN11) predicted sensitivity to DNA-damaging alkylating agents and PARP inhibitors [103–106]. The *SLFN11* gene was observed to be under epigenetic control and methylation of its promoter in ctDNA predicted resistance to therapy in ovarian cancer with the prior mentioned agents [107]. Hypermethylation of the promoter of the *MLH1* gene represented a principal epigenetic mechanism that led to the MSI-H/dMMR state. Tumors with deficient mismatch repair possessed a higher burden of somatic mutations, higher infiltrating T-cell counts, and showed increased PD-L1 positivity—all these predispose MSI-H/dMMR cancers to respond to immune checkpoint inhibitor (ICI) therapy [108,109]. Wang et al. [110] described a liquid biopsy-based approach to the detection of *MLH1* promoter methylation in CRC using methylation-sensitive restriction enzyme PCR. This assay had an AUROC value of 0.965 (95% CI: 0.94–0.99). The sensitivity and specificity of the assay were 78% and 100%, respectively (95% CI: 0.45–0.95).

Beyond individual genes, methylation-profile-based scoring also predicted anti-cancer therapy response. In an exploratory study from the phase 2 SWOG S1314 clinical trial, pre-treatment cfDNA from 72 patients was used to generate a classifier methylation-based response score (mR-score), which could predict pathologic response to neoadjuvant chemotherapy in operable, muscle-invasive bladder cancer (MIBC). The model demonstrated an AUROC of 0.636 (95% CI: 0.498–0.773) for baseline samples. Moreover, mR-scores correlated with pathologic response with complete responders demonstrating the lowest scores in the cohort and non-responders showing the highest mR-scores. Notably, this same classifier showed better performance when applied to on-treatment samples (AUROC of 0.720, 95% CI: 0.582–0.857) [111]. Methylation-profile-based scoring was also used to predict therapy response in ICI. In a sub-study of the INSPIRE trial (NCT0264436), ctDNA methylation kinetics in 51 heavily pre-treated patients were used to predict response to the anti-PD1 monoclonal antibody Pembrolizumab. A decrease in both ctDNA fraction and cfDNA methylation (cfMEDIP score) predicted the best response to Pembrolizumab (12mo OS 87.5%, 95% CI: 72.7–100%). Meanwhile, a decrease in either ctDNA fraction or cfMeDIP score also showed benefit (12mo OS 70%, 95% CI: 46.6–100% and 12mo OS 62.5%, 95% CI: 36.5–100%, respectively) compared to patients that demonstrated an increase in both parameters following ICI (12mo OS 29.4%, 95% CI: 14.1–61.4%) [112,113]. MethylCIBERSORT utilized methylation profiles to predict the TME milieu and distinguished between immune hot and immune cold tumors. This approach was further validated across multiple tumor sites including head and neck carcinomas, CRC, glioma, and melanoma [114–117]. Differential Methylation Analysis for Immune Cell Estimation (DIMEimmune) could be another approach to determine tumor-infiltrating lymphocytes (TILs) scores as well as estimation of

CD8$^+$ and CD4$^+$ T-cell abundance. This technique used bulk methylation data and was less reliant on cell line data than the former approach [118]. Studies have also reported on the use of "methylation scores" to predict response to ICI therapy. Liu et al. [119] developed a 5mC score based on the expression of twenty-one 5mC regulators in patients with lung adenocarcinoma. The 5mC score was shown to predict ICI response and prognosis. The authors reported that a low 5mC score corresponded with more immune cell infiltration and improved response to ICI therapy. A similar approach was described by He et al. [120] for developing a predictive 5mC score in lung squamous cell carcinoma treated with ICI. Their cohort patients with a high 5mC score also showed diminished sensitivity to immunotherapy in contrast to those with a low 5mC score.

### 2.3.4. Minimal Residual Disease

Effective oncologic therapy reduces tumor burden by >2–3 log$_{10}$ which corresponds to a tumor kill of >99%. Unfortunately, malignant cells may persist in patients who achieve exquisite treatment responses and represent niduses for disease relapse. These recalcitrant cancer cells, referred to as minimal residual disease (MRD), remain a key challenge in the treatment of cancer [121]. To guarantee the durability of remission following curative-intent therapy, increasingly sensitive methods for post-treatment surveillance are necessary [58]. Knowing that recurrent tumors detectable by conventional imaging contain >10$^6$ cancer cells, the likelihood of successful treatment becomes less likely once it is demonstrable by diagnostic scans [122]. Detection of ctDNA by liquid biopsies of different biofluids, such as plasma and urine, represents an avenue for earlier detection of recurrent disease thereby facilitating cancer interception, earlier intervention, and more optimal treatment outcomes [123–128].

Van Zogchel et al. [125] demonstrated that the detection of hypermethylated *RASSF1A* in the plasma of cohort patients with different solid tumors (neuroblastoma, renal tumors, rhabdomyosarcoma, or Hodgkin lymphoma) predicted negative outcomes following chemotherapy. In this study, the performance of hypermethylated *RASSF1A* was best when combined with the detection of RASSF1A mRNA in patient bone marrow aspirates ($p = 0.046$). The results of this small cohort ($n = 96$) were to be confirmed in the larger, prospective, multicenter SIOPEN HR-2 study (NCT04221035). Mo et al. [127] investigated the utility of detecting MRD via ctDNA using six DNA methylation markers, which included the well-studied *SEPT9*, in post-treatment blood samples from patients with resectable CRC. Of 255 patients with available plasma samples, 59 were ctDNA-positive. In this group, the recurrence risk was significantly higher than that of their counterparts (HR 17.5; 95% CI: 8.9–34.4; $p < 0.001$). Notably, the optimal timing of measurement of ctDNA was ≤1 month after surgery [126–128]. The prospective MEDAL study investigated the feasibility of methylation-based postoperative surveillance in patients with resected NSCLC (n = 195). The ctDNA methylation-based MRD model recapitulated perioperative ctDNA mutation dynamics. Comparing the two methods, ctDNA methylation had better sensitivity than ctDNA mutations (90.9% vs. 45.5%) but the latter was more specific (90.4% vs. 67.3%). Moreover, DFS was shorter for those with high methylation-based MRD scores (HR 15.32; 95% CI: 1.96–119.76; $p < 0.001$) and the average lead time before confirmation of disease recurrence by clinical imaging was 137 days for those with ctDNA determined by mutation status vs. 303 days for those with high methylation-based MRD scores. This represented an earlier window of intervention afforded by the latter approach [129]. It is worth mentioning that some MRD assays have become available for clinical use. A tumor-uninformed assay designed to integrate mutation and aberrant methylation detection for monitoring MRD was validated in CRC and breast cancer (Table 1) [92,93]. At the time of writing, cfMeDIP-seq was being developed for MRD testing and would offer the advantage of requiring low DNA inputs (5–10 ng), making it an attractive platform for monitoring treatment response [71,76,130].

### 3. Challenges and Future Directions

*3.1. DNA Methylation in Oncology*

Samples for DNA methylation profiling are subject to the same requirement of quality as other biological materials for analysis. Specimens with low tumor content, inadequately prepared samples, or old archival material may yield suboptimal DNA quality that can confound discrimination between tumor and non-tumor material [131]. The variety of platforms for studying DNA methylation offers distinct advantages in terms of ease of use, assay resolution, and coverage. However, it has to be pointed out that each method also has its inherent limitations. For example, bisulfite conversion can result in sample degradation, while antibody performance can present a challenge for methylated DNA enrichment techniques [73–75]. Furthermore, although DNA methylation provides valuable information, this may need to be incorporated with additional omics data to provide greater insight into tumor biology [71,113]. To this end, to leverage the expansive omics data available, the use of machine learning algorithms that enable the integration of multi-omics data for general or task-specific uses is essential and is an actively growing field of study [132]. Importantly, information gleaned from multi-omics analyses can pave the way for biomarker discovery. Such applications will ultimately need to be validated using appropriate prospective clinical studies to confirm their value [133].

*3.2. Liquid Biopsy in Oncology*

The use of tumor cfDNA is not without its limitations. First, there is variability in the timing and amount of DNA shed into the circulation [58]. The challenge is compounded by the minuscule amount of nucleic acid that can be obtained from the blood in comparison to material from conventional biopsy techniques. Also, circulating DNA has a short half-life in the blood of a few minutes to up to two hours [134]. Second, cfDNA is shed by both tumor and non-tumor tissue and this makes it difficult to parse out tumor-specific signals [68,69,75]. Third, since ctDNA may reveal greater molecular heterogeneity than can be captured by a needle biopsy obtained from a single tumor site, defining the gold standard to use for determining the specificity and sensitivity of assays may be challenging [135].

The earlier constraints arising from the biology of ctDNA can be overcome by performing careful and rapid isolation of cfDNA from patient blood to avoid lysis of normal blood cells resulting in contamination of samples with non-cancer cfDNA [56]. Alternatively, cfDNA stabilizing tubes can be used to store samples that are not amenable to immediate processing [56,135]. Leveraging advances in next-generation sequencing and computational methods can distinguish cancer-associated methylation signatures from non-cancer signals during downstream analysis [56,57,67]. Employing multiple patient sample sources in addition to plasma will increase the sensitivity of ctDNA testing but this needs to be balanced against the risk of increasing false-positive cancer signals during testing [135].

*3.3. ctDNA Methylation as a Biomarker in Oncology*

It is important to note that for the myriad methods described here for ctDNA methylation-based biomarkers, questions regarding clinical validity and utility remain. This is particularly true in the case of MRD detection where validation of the positive and negative predictive value of assays are few [136,137]. As many of the approaches described are tailored for specific cancers using retrospective data, their broader utility needs to be prospectively tested in the context of clinical trials. Other challenges with 'bespoke' approaches relate to generalizability—is an assay robust enough to be used in more than a few clinical scenarios? Practicability is a closely related issue, which represents a key stumbling block for translating technologies from bench to clinic [136,138].

Looking forward, work on developing ctDNA methylation-based MCED illustrates the progress being made at developing tumor-uninformed assays that can perform reliably in the clinic. At present only four cancers: breast, cervix, colon, and lung have screening tests with evidence supporting a cost-effective reduction in mortality [81]. The development of pan-cancer screening tests based on ctDNA methylation is, therefore, a promising

avenue with potential benefit for patients as it provides opportunities for the prevention of death or disability arising from cancer. However, balancing this with false positives or detection of diseases that do not have any effective treatment options will be necessary. The results of the ongoing validation trials for the GRAIL and cfMeDIP-seq-based assay are eagerly anticipated.

Meanwhile, the success of liquid biopsy-based assays in the clinic is related to their ability to complement conventional tests to detect MRD during treatment response and post-treatment monitoring. Efforts to push the limit of detection (LOD) of ctDNA ever lower are central to arriving at an effective test to monitor MRD that performs superior to current imaging and serum-based markers. The analytical limits of ctDNA assays are typically quantified in terms of variant allele frequency (VAF)—the percentage of sequencing reads with tumor-specific mutations in relation to the total number of sequencing reads overlapping the same genomic loci [139]. Advances in next-generation sequencing (NGS) are crucial in driving down the LOD of ctDNA tests. The seminal hybrid capture-based NGS platform CAPP-seq allowed the detection of VAFs down to ~0.02% with 96% specificity and 85% sensitivity [140]. Advancements in this approach pushed the limit of detection much lower, to ~0.003% with 100% specificity and 94% sensitivity [64]. Furthermore, other studies showed that ctDNA fragmentation and methylation were biologically interrelated and can complement mutation-based assays in cancer detection [64,71,141].

Finally, as ctDNA is present in different biofluids, sampling strategies that exploit the spatial relationship between tumor site and biofluid type can further increase the sensitivity of ctDNA detection. Especially important in the case of low disease burden settings, biofluids sampled proximal to a tumor site (e.g., urine for urothelial carcinoma) [142] can be more enriched for ctDNA than plasma [64]. Combining these approaches creates opportunities for establishing highly sensitive ctDNA-based MCED and MRD tests for use in the clinic.

## 4. Conclusions

Methylation influences carcinogenesis through the epigenetic control of gene expression and maintenance of genomic integrity. Its roles in the regulation of the noncoding genome and its influence on the TME are emerging areas of study in cancer biology. Due to the highly conserved nature of cancer-specific methylation, its detection in cfDNA using liquid biopsies has been demonstrated and constitutes an area of great interest in biomarker research in terms of diagnosis, treatment planning, and response evaluation. Notably, several key challenges need to be addressed before cfDNA-based methylation biomarkers become integrated into clinical practice. However, these challenges do not preclude the development of ctDNA methylation-based biomarkers, as shown by the increasing number of regulator-approved targeted and tumor-uninformed biomarker assays that have entered the clinic in recent years.

**Author Contributions:** Summarized below are the author contributions to this review. Conceptualization, D.B.S., S.U.H. and B.H.L.; Methodology, D.B.S., S.U.H. and B.H.L.; Writing—Original Draft Preparation, D.B.S.; Writing—Review and Editing, D.B.S., S.U.H. and B.H.L.; Supervision, B.H.L. All authors have read and agreed to the published version of the manuscript.

**Funding:** Research in the B.H. Lok laboratory is supported by the Terry Fox Research Institute, Canada Foundation for Innovation, Cancer Research Society, Canadian Institutes of Health Research, National Institute of Health/National Cancer Institute (U01CA253383), Clinical and Translational Science Center at Weill Cornell Medical Center, MSKCC (UL1TR00457). D.B.S. is supported by the Institute of Medical Science. S.U.H. is supported by the Institute of Medical Science, Canadian Institutes of Health Research, and The Strategic Training in Transdisciplinary Radiation Science for the 21st Century (STARS21).

**Acknowledgments:** The authors wish to acknowledge the invaluable contribution of Vivek Philip, and Janice Li in editing the manuscript. We also thank the other members of the Lok lab at the Princess Margaret Cancer Centre for their support in the preparation of this work.

**Conflicts of Interest:** B.H. Lok reports grants from Pfizer and grants, personal fees, and non-financial support from AstraZeneca outside the submitted work. The other authors declare no conflict of interest in relation to this review.

## Appendix A. Narrative Review Search Strategy

In preparing this narrative review, the authors conducted a literature search using the following search terms: "DNA methylation", "5 methylcytosine", "5-mc", "methylcytosine", "cell free DNA", "cfDNA", "circulating tumor DNA", "ctDNA", "cancer", "oncology", "liquid biopsy", "biomarker", "plasma", "prognostic biomarker", "predictive biomarker", "minimal residual disease", "MRD". Databases utilized included PubMed, Europe PMC, LILACS, Elsevier Science Direct, Web of Science, ClinicalTrials.gov, and Google Scholar. In addition, conference proceedings were perused for relevant abstracts. The references of included studies were hand searched for additional relevant papers. No limits were placed on the date of publication. Included studies consisted of clinical trials, descriptive cohort studies, and preclinical studies related to the theme of the review. Abstracts published in conference proceedings were also considered for inclusion if these reported pertinent information. Studies determined by the authors to be outside the scope and theme of the review were excluded. Only papers published in English were considered for inclusion. Figure A1 summarizes the search strategy and results.

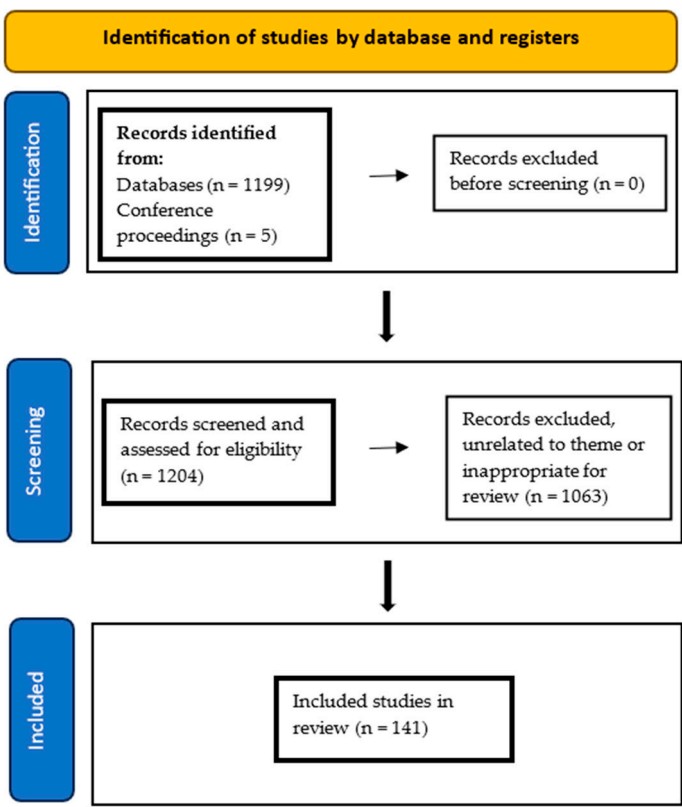

**Figure A1.** Flow diagram of search strategy of this narrative review.

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
