# Peer review of "Plasma Cell-Free Tumor Methylome as a Biomarker in Solid Tumors: Biology and Applications"

_curroncol, doi:10.3390/curroncol31010033_

Round 1

Reviewer 1 Report

Comments and Suggestions for Authors

The review analyzes the possibilities of the use of DNA methylation as a biomarker in oncology. The review is interesting, well written and well organized. Mi major concern is that introduction is too short. A good introduction should include a description of the importance of DNA methylation, main mechanism of action, CpG island, role of methylation in cancer, objetives of the study... Indeed a short section with DNA methylation biomarkers in other fluids (no plasma) would be informative. 

Minor points:

- Authors should add a tittle oin section 3.1

Author Response

The review analyzes the possibilities of the use of DNA methylation as a biomarker in oncology. The review is interesting, well written and well organized. Mi major concern is that introduction is too short. A good introduction should include a description of the importance of DNA methylation, main mechanism of action, CpG island, role of methylation in cancer, objetives of the study... Indeed a short section with DNA methylation biomarkers in other fluids (no plasma) would be informative. 

We thank the reviewer for their kind comments. We recognized that the introduction was short and have added to this section. Furthermore, we moved the methods section to appendix A to improve flow into section 2.

Methylated cell free DNA can be obtained from biofluids other than plasma and this has been discussed briefly in section 2.2 and 2.3, however we have been keen to focus on the use of plasma cfDNA as biomarker in cancer as this is what he hoped to spotlight in this review.

Minor points:

- Authors should add a tittle oin section 3.1

Thank you for pointing this out. We have added a title to this section.

Reviewer 2 Report

Comments and Suggestions for Authors

This is a well written manuscript, i have several minor issues that should be addressed:

1. Please make sure to explain all abbreviations when first used (TSSs line 54, page 2)

2. I do not think that explaining abbreviations in subtitles of individual parts of the manuscript is good

3. Line 34 - please rephrase ..methylation can be detected...BY liquid biopsies.....

4. Why do the authors refer to cell free DNA as 'nuclear'?

5. liquid biopsies are not always non-invasive, that would be true only for urine samples. Generally, they are minimally invasive. Even for the purposes of this review, the authors should not use the term only for peripheral blood. It is also not only the access to the tumor but also ethical issues of sampling too many times

6. it would make more sense to change the title of the review to emphasize that the review is on solid tumors only...and it would make more sense to review only either pediatric or adult tumors....

Comments on the Quality of English Language

I did not detect any major issues

Author Response

This is a well written manuscript, i have several minor issues that should be addressed:

We thank the reviewer for taking the time to go over the manuscript and for their kind comments. We wish to be able to address them below.

  1. Please make sure to explain all abbreviations when first used (TSSs line 54, page 2)

Thank you for pointing this out. We have defined the abbreviation (transcription start site) in the text.

  1. I do not think that explaining abbreviations in subtitles of individual parts of the manuscript is good

We appreciate this point and agree. We have rectified section headings in the text accordingly.

  1. Line 34 - please rephrase ..methylation can be detected...BY liquid biopsies...

We have rephrased the sentence as follows: ‘Liquid biopsies can be used to obtain cell-free DNA from different biofluids such as cerebrospinal fluid, pleural effusion, urine, and plasma.’

  1. Why do the authors refer to cell free DNA as 'nuclear'?

We have removed the reference to ‘nuclear’ in the text.

  1. liquid biopsies are not always non-invasive, that would be true only for urine samples. Generally, they are minimally invasive. Even for the purposes of this review, the authors should not use the term only for peripheral blood. It is also not only the access to the tumor but also ethical issues of sampling too many times

The reviewer makes an excellent point. We removed the qualifying statement alluded to. Additionally, we have taken this into account and revised the text as follows: 'Liquid biopsies using standard venipuncture afford several distinct advantages over conventional tissue sampling. First, their non minimally invasive nature allows evaluation of disease particularly in cases where tumor access is difficult, or unsafe, or ethically challenging especially for repeated longitudinal sampling.'

  1. It would make more sense to change the title of the review to emphasize that the review is on solid tumors only...and it would make more sense to review only either pediatric or adult tumors....

Thank you for this comment. We have revised the title as follows: “Plasma Cell Free Tumor Methylome as a Biomarker in Solid Tumors: Biology and Applications”.

We hope that this better captures the topic of the review.

Reviewer 3 Report

Comments and Suggestions for Authors

-          1. Flow diagram of your literature research will be very helpful and informative. Please add a flow diagram of your literature research and explain the exclusion criteria you used.

-          2. Please define TSSs (line 54)

-          3. Hypermethylation occurs on DNA. What do you mean when you say “Hypermethylation of transcriptional regulators of tumor suppressor genes (TSGs) results in their silencing and the abrogation of their anti-tumor function” (line 61-62). Where occurs the hypermethylation? On transcription factors? In this case, it is a protein methylation.

-          4. Line 155-156. What do you mean?” Demethylation of promoters of cytokines and transcription factors by TET2 promotes gene expression and directs differentiation into CD4+ T-cell subtypes”.  Is the promoter demethylated? Are the transcription factors, demethylated? And do you mean the action of TET as protein? Or do you that TET2 gene is demethylated?

-          5. In line 174 you mention “….DNMT3b and DNMT1 are….” I assume you are referring to proteins in both cases. Why do you write with italics?

-          6. No reference of Fig 1 in the text.

-          7. Line 277. “such as SEPT9 [76,77], SHOX2 [78], PTGER4 [78], and SDC2 [79].” Do you refer to genes (promoter region, transcriptional site) or to the proteins?

-          8. Line 316 “treatment naïve patients”?

 Additionally:

-          9. It is suggested to refer to the use of urine samples, to detect DNA methylation alteration as biomarkers for early cancer diagnosis.

-          10. Moreover, it is suggested to refer to PMID: 35838992, as an example for the prognostic value of methylation for metastatic cancer.

-          11. Beside DNA methylation, there is also protein methylation on arginine and lysine residues. Therefor it is strongly suggested to use the term “DNA methylation” and not only “methylation”.

Comments on the Quality of English Language

 1. The authors refer at previous studies that have been performed in the past. It is suggested to use the simple past tense that shows they are referring to a study that has already published.

2. Please note that we use italics when we refer to genes. When we refer to proteins, we DO NOT USE italics. Please correct accordingly whole manuscript.

Author Response

We thank the reviewer for taking the time to go over the manuscript and for their insightful comments. We wish to be able to address them below.

1. Flow diagram of your literature research will be very helpful and informative. Please add a flow diagram of your literature research and explain the exclusion criteria you used.

      Thank you for your suggestion. While the work is a narrative review, and unlike a systematic review, does not require a (PRISMA) flow diagram. We expanded the methods section and added a flow diagram as requested. To improve the flow for readers, we moved this section and the accompanying figure to the manuscript appendix (A).

2. Please define TSSs (line 54)

Thank you for pointing this out. We have defined the abbreviation (transcription start site) in the text.

3. Hypermethylation occurs on DNA. What do you mean when you say “Hypermethylation of transcriptional regulators of tumor suppressor genes (TSGs) results in their silencing and the abrogation of their anti-tumor function” (line 61-62). Where occurs the hypermethylation? On transcription factors? In this case, it is a protein methylation.

      We appreciate the reviewer’s comments. The text has been revised to improve clarity as follows: ‘Hypermethylation of transcriptional regulatory regions of tumor suppressor genes (TSGs) results in their silencing and the abrogation of their anti-tumor function...’

4. Line 155-156. What do you mean?” Demethylation of promoters of cytokines and transcription factors by TET2 promotes gene expression and directs differentiation into CD4+ T-cell subtypes”.  Is the promoter demethylated? Are the transcription factors, demethylated? And do you mean the action of TET as protein? Or do you that TET2 gene is demethylated?

      We thank the reviewer for their comment. The text has been revised to improve clarity as follows: ‘To illustrate, TET2 demethylates promoters of genes encoding cytokines and transcription factors that influence CD4+ T-cell fate. Expression of the cytokine IL-4 favors differentiation into Th2 cells while expression of the transcriptional regulator FOXP3 leads to differentiation into tissue regulatory T-cells (Treg)...’

5. In line 174 you mention “…. DNMT3b and DNMT1 are….” I assume you are referring to proteins in both cases. Why do you write with italics?

      Thank you for this accurate observation. Both refer to proteins and DNMT1 is no longer italicized in the text.

6. No reference of Fig 1 in the text.

      The reviewer is correct. We have added the appropriate reference prior to the figure.

7. Line 277. “such as SEPT9 [76,77], SHOX2 [78], PTGER4 [78], and SDC2 [79].” Do you refer to genes (promoter region, transcriptional site) or to the proteins?

      Thank you for pointing this out. The text has been revised for clarity and now reads as follows: ‘Assays that employ ctDNA methylation for this application vary in their approaches. Some use targeted detection of methylated sites in specific genes, such as SEPT9, SHOX2, PTGER4, and SDC2 while others profile more extensive swathes of the genome...’

8. Line 316 “treatment naïve patients”?

      Thank you for this question. We had intended to refer to untreated patients. To ensure clarity we have rephrased accordingly.

 Additionally:

9. It is suggested to refer to the use of urine samples, to detect DNA methylation alteration as biomarkers for early cancer diagnosis.

Methylated cell free DNA can be obtained from biofluids other than plasma and this has been discussed briefly in section 2.2, however we have been keen to focus on the use of plasma cfDNA as biomarker in cancer as this is what he hoped to spotlight in this review. Use of urine for cancer detection has been alluded to in section 3.3 in relation to selecting proximal biofluids for cfDNA sampling.

10. Moreover, it is suggested to refer to PMID: 35838992, as an example for the prognostic value of methylation for metastatic cancer.

      We thank the reviewer for this suggestion. This paper further supports the prognostic value of DNA methylation in cancer. Accordingly, we have cited this study (ref. 7) in the manuscript.

11. Beside DNA methylation, there is also protein methylation on arginine and lysine residues. Therefor it is strongly suggested to use the term “DNA methylation” and not only “methylation”.

This is an excellent point. We have ensured that the text reflects this suggestion.

Comments on the Quality of English Language

  1. The authors refer at previous studies that have been performed in the past. It is suggested to use the simple past tense that shows they are referring to a study that has already published.

The reviewer is once again correct. We have revised all errant sections.

  1. Please note that we use italicswhen we refer to genes. When we refer to proteins, we DO NOT USE italics. Please correct accordingly whole

We agree with the reviewer and thank them for pointing this out. We have revised the text accordingly.

Round 2

Reviewer 1 Report

Comments and Suggestions for Authors

Thge manuscript has been improved and it is suitable for publication

Reviewer 3 Report

Comments and Suggestions for Authors

Thank you for revising the manuscript and for your responses.